# *Arabidopsis* LSH10 transcription factor and OTLD1 histone deubiquitinase interact and transcriptionally regulate the same target genes

Mi Sa Vo Phan [1✉], Ido Keren[1], Phu Tri Tran[1], Moshe Lapidot[2] & Vitaly Citovsky [1]

Histone ubiquitylation/deubiquitylation plays a major role in the epigenetic regulation of gene expression. In plants, OTLD1, a member of the ovarian tumor (OTU) deubiquitinase family, deubiquitylates histone 2B and represses the expression of genes involved in growth, cell expansion, and hormone signaling. OTLD1 lacks the intrinsic ability to bind DNA. How OTLD1, as well as most other known plant histone deubiquitinases, recognizes its target genes remains unknown. Here, we show that *Arabidopsis* transcription factor LSH10, a member of the ALOG protein family, interacts with OTLD1 in living plant cells. Loss-of-function LSH10 mutations relieve the OTLD1-promoted transcriptional repression of the target genes, resulting in their elevated expression, whereas recovery of the LSH10 function results in down-regulated transcription of the same genes. We show that LSH10 associates with the target gene chromatin as well as with DNA sequences in the promoter regions of the target genes. Furthermore, without LSH10, the degree of H2B monoubiquitylation in the target promoter chromatin increases. Hence, our data suggest that OTLD1-LSH10 acts as a co-repressor complex potentially representing a general mechanism for the specific function of plant histone deubiquitinases at their target chromatin.

[1] Department of Biochemistry and Cell Biology, State University of New York, Stony Brook, NY 11794-5215, USA. [2] Department of Vegetable Research, Institute of Plant Sciences, Agricultural Research Organization, The Volcani Institute, 68 HaMaccabim RdP.O.B 15159 Rishon LeZion 7505101, Israel. ✉email: vpmisa@gmail.com

Growing in a dynamic environment, plants must deal with numerous biotic and abiotic stress sources. Histone modifications that underly epigenetic regulation of gene expression help plants to adapt to environmental changes and serve as an environmental memory of the transcription[1]. Histone modifications not only regulate gene expression in response to diverse environmental signals such as stress[2], pathogen attack[3], temperature[4,5], and light[6] but play an important role in plant development and morphogenesis [reviewed in[4,7–10]].

Covalent modifications of histones can activate or repress transcription by altering the chromatin structure[11]. However, because histone-modifying enzymes often do not possess a DNA binding ability, they are not sufficient to trigger transcriptional regulation of the specific target genes. To achieve this regulation, histone-modifying enzymes are thought to function in complexes with transcription factors that contain DNA-binding domains. This DNA binding capacity is thought to direct histone-modifying enzymes, as components of the histone modifier-transcription factor complexes, to the target promoters. Indeed, in plants, different transcription factors have been shown to act in concert with such diverse histone modifiers as histone methyltransferases[12], histone acetyltransferases[13], histone demethylases[14], and Polycomb repressive complexes that promote histone trimethylation and monoubiquitylation[15]. Yet, whether transcription factors can also facilitate the function of plant histone deubiquitinases, one of the major types of histone-modifying enzymes, remains unknown.

OTLD1, which belongs to the ovarian tumor (OTU) deubiquitinase family, represents one of the few characterized plant histone deubiquitinases[16,17]. Our previous studies have shown that OTLD1 mainly functions as a transcriptional co-repressor (although, at least in one case, OTLD1 facilitated transcriptional activation of its target gene) by associating with the target chromatin and deubiquitylating histone 2B (H2B) at the occupied regions, thereby promoting the erasing or writing of euchromatic histone marks[18,19]. Specifically, monoubiquitylation of H2B represents one of the molecular reactions that mediate epigenetic regulation of plant physiology from leaf and root growth to seed dormancy to circadian clock to photomorphogenesis[18–24]. The erasure of the monoubiquityl marks from the histone molecules by histone deubiquitinases facilitates changes in histone methylation and acetylation, bringing about transcriptional repression or activation of the corresponding genes[25,26]. Consistent with this role in epigenetic regulation of transcription, OTLD1 can interact and crosstalk with the histone demethylase KDM1C to coordinate histone modification and transcriptional regulation of the target genes[27,28]. But how is OTLD1 directed to the target gene promoters? To address this question, we endeavored to identify a putative DNA-binding protein that recognizes OTLD1 and targets it to the sites of its biological function.

Here, we report that LSH10, a member of the ALOG (Arabidopsis LSH1 and Oryza G1) protein family, interacts with OTLD1 and participates in transcriptional repression of the OTLD1 target genes *OSR2, WUS, ABI5,* and *ARL.* The expression of these genes was elevated in the *LSH10* loss-of-function mutants and suppressed in the gain-of-function lines. Specific interaction between LSH10 and OTLD1 was demonstrated by two independent fluorescence imaging techniques in living plant cells. The binding of LSH10 to the chromatin and DNA sequences of the *OSR2, WUS, ABI5,* and *ARL* promoters was also demonstrated. The *LSH10* loss-of-function plants also exhibited enhanced H2B monoubiquitylation in the target promoter chromatin. Taken together, these observations suggest that LSH10 functions as a transcription factor that interacts with the OTLD1 co-repressor for transcriptional control of a common set of target genes.

## Results

**LSH10 is a nuclear protein that interacts with OTLD1.** To gain better insight into possible mechanisms by which plant histone deubiquitinases reach their target chromatin, we searched for proteins that interact with OTLD1. A truncated OTLD1 was used as bait to screen the Arabidopsis yeast two-hybrid protein interaction library[29], and two independent clones encoding the LSH10 protein were identified as a putative interaction partner of OTLD1 (Supplementary Fig. 1). We then examined the sub-cellular localization of LSH10, which was tagged with CFP and transiently expressed in *N. benthamiana* leaf tissues together with a free YFP reporter that partitions between the cell cytoplasm and the nucleus, conveniently visualizing and identifying both sub-cellular compartments. Figure 1a shows that LSH10-CFP accumulated in the cell nucleus of the cells whereas, as expected, the free YFP fluorescence was nucleocytoplasmic. This nuclear localization of LSH10 is consistent with its interaction with OTLD1, a histone deubiquitinase that functions in the cell nucleus[27].

Next, the physical interaction of LSH10 with OTLD1 was studied *in planta* using two independent approaches, bimolecular fluorescence complementation (BiFC) and fluorescence resonance energy transfer (FRET). BiFC experiments shown in Fig. 1b detected a strong fluorescent signal of the reconstituted

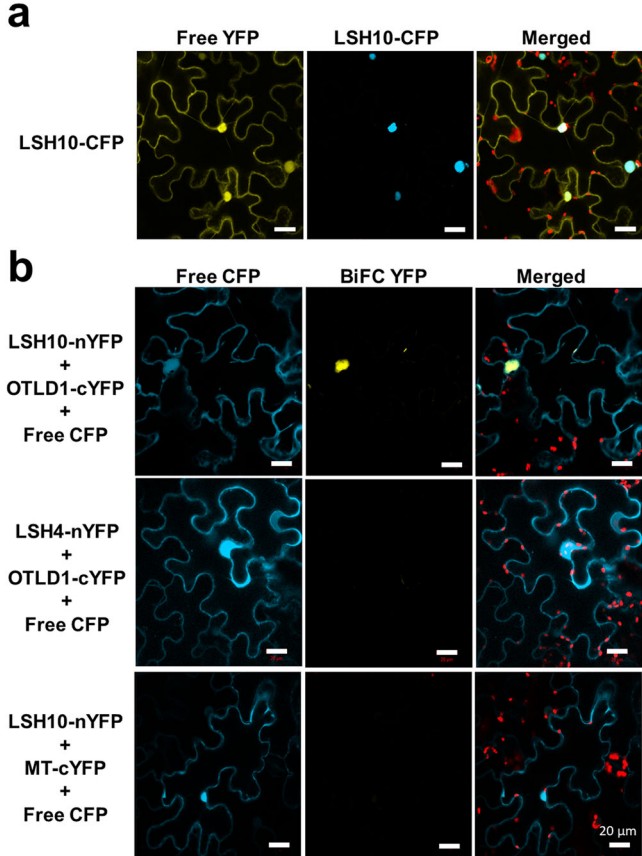

**Fig. 1 LSH10 subcellular localization, and specific interaction of LSH10 with OTLD1 *in planta* detected by BiFC. a** Nuclear localization of LSH10. The LSH10 protein tagged with CFP (LSH10-CFP) and free YFP were transiently co-expressed in *N. benthamiana* leaves. **b** The BiFC assay of the interaction between LSH10 and OTLD1. The indicated combinations of proteins tagged with nYFP and cYFP were transiently co-expressed in *N. benthamiana* leaves. CFP signal is in cyan; YFP signal is in yellow; merged CFP and YFP signals are in cyan-yellow, and chlorophyll autofluorescence is in red. Scale bars = 20 μm. All images are single confocal sections.

YFP in the cells co-expressing both LSH10 and OTLD1, indicating protein interaction. This interaction was specific as no BiFC signal was observed in the cells co-expressing either OTLD1 and LSH4, a homolog of LSH10, or LSH10 and an unrelated plant viral protein MT (Fig. 1b). Furthermore, the interacting proteins colocalized with the nuclear portion of the coexpressed free YFP reporter, indicating that the OTLD1-LSH10 complexes were located in the cell nucleus, the expected subcellular site of their function (Fig. 1b). Our FRET experiments—using LSH10 tagged with GFP as donor fluorophore and OTLD1 tagged with RFP as acceptor fluorophore—confirmed and extended the BiFC findings. We used two variations of the FRET method, sensitized emission (SE-FRET) and acceptor bleaching (AB-FRET)[30]. In SE-FRET, protein interaction results in the transfer of the excited state energy from the GFP donor to the RFP acceptor without emitting a photon, producing the fluorescent signal with an emission spectrum similar to that of the acceptor. AB-FRET, on the other hand, detects and quantifies protein interaction from increased emission of the GFP donor when the RFP acceptor is irreversibly inactivated by photobleaching. Figure 2a summarizes the results of the SE-FRET experiments, in which the cell nuclei were simultaneously recorded in all three, i.e., donor GFP, acceptor RFP, and SE-FRET, channels and used to generate images of SE-FRET efficiency illustrated in a rainbow pseudo-color. This color scale, i.e., transition from blue to red, indicates an increase in FRET efficiency from 0 to 100%, which corresponds to the degree of protein-protein proximity during the interaction. The SE-FRET signal observed in the cell nuclei following the coexpression of LSH10 and OTLD1 was comparable to that generated in positive control experiments which expressed the translational acceptor-donor RFP-GFP fusion. Negative controls, i.e., coexpression of OTLD1-RFP with LSH4-GFP or free RFP with LSH10-GFP, produced no SE-FRET signal (Fig. 2a). The FRET data were quantified using AB-FRET (Fig. 2b, c) by recording the cell nuclei in the donor GFP channel before and after RFP photobleaching and displayed in pseudo-color to visualize the change in GFP fluorescence. Figure 2b shows that photobleaching of the RFP acceptor completely blocked its fluorescence in all protein coexpression combinations tested. Following this photobleaching, two protein combinations showed an increase in the GFP donor signal, i.e., LSH10-GFP coexpressed with OTLD1-RFP and the RFP-GFP fusion positive control. In contrast, the negative controls, i.e., LSH4-GFP coexpressed with OTLD1-RFP and LSH10-GFP coexpressed with free RFP, elicited no increase in the GFP fluorescence (Fig. 2b). Quantification of these data demonstrated that the increase in the donor fluorescence (%AB-FRET) of 13% observed following LSH10-OTLD1 coexpression was statistically significant and overall comparable to the maximal %AB-FRET of 30% achieved with RFP-GFP. Both negative controls displayed no increase in donor fluorescence (Fig. 2c). Collectively, the data in Fig. 2 indicate that LSH10 interacts with OTLD1 within living plant cells, that the interacting proteins accumulate in the cell nucleus, and that, in the LSH10-OTLD1 complex, the proteins are within <10 nm from each other, the effective range of protein interactions detected by FRET[31].

For the LSH10-OTLD1 interaction to be biologically meaningful, both genes should be expressed at least in some of the same plant tissues at the same time. Thus, we examined the expression pattern of the endogenous *LSH10* and *OTLD1* genes in different organs systematically throughout the plant, i.e., rosette and cauline leaves, stems, flowers, and roots of wild-type plants using reverse transcription-quantitative RT-PCR (RT-qPCR) analysis. Figure 3 shows that both genes were expressed in all tested tissues, suggesting the availability of their protein products for functional interaction during transcriptional regulation of their target genes. Obviously, the expression levels in different tissues varied. For example, *LSH10* was expressed most prominently in roots, less in stems and rosette leaves, and at the lowest relative levels in flowers and cauline leaves (Fig. 3). As in earlier observations[19], *OTLD1* was expressed at the highest level in flowers and lower levels in all other tested tissues (Fig. 3). Overall, regardless of the exact degree of expression, *LSH10* and *OTLD1* transcripts were detected all tested tissues, suggesting that their protein products are available for functional interaction during transcriptional regulation of their target genes and that the interplay between the relative expression levels of *LSH10* and *OTLD1* may contribute to the overall regulation pattern of these target genes.

**LSH10 has the structural features of a transcription factor**. LSH10 is a 177-amino acid residue protein (Supplementary Fig. 2) encoded by the Arabidopsis At2G42610 gene. It belongs to a 10-member family of ALOG (Arabidopsis LSH1 and Oryza G1) protein family in Arabidopsis (Supplementary Fig. 3a) as well as in numerous other dicotyledonous plant species (Supplementary Fig. 3b). The members of this family, many of which remain uncharacterized, carry a highly conserved ALOG domain (also known as DOMAIN OF UNKNOWN FUNCTION 640 / DUF640) located in the center of the protein molecule. This domain is composed of 4 all-α helices, a zinc ribbon insert structure, and a nuclear localization signal (NLS) (Supplementary Fig. 2). ALOG is predicted to act as a DNA binding domain and belongs to the tyrosine recombinase/phage integrase N-terminal DBD superfamily[32], in which the ALOG domain members, unlike the tyrosine recombinase members, contain a conserved zinc ribbon insert located between helices 2 and 3 with highly conserved positively charged residues at its N-terminus and the "HxxxC" and "CxC" motifs. This region can provide additional molecular contacts unique to the ALOG domain to participate in binding to DNA. The conserved ALOG sequences (Supplementary Fig. 3) and our prediction of the DNA-binding amino acid residues using three methods, DRNApred[33], DP-Bind[34], and DISPLAR[35], indicate that LSH10 may associate with DNA via hydrogen bonding and ionic interactions with multiple conserved solvent-accessible basic residues[36] in helix-1, helix-3, helix-4, and in the zinc ribbon or with a conserved acidic residue[36] in helix-1, whereas the conserved hydrophobic residues in all four helices likely stabilize the core tetra-helical fold[37] in the target DNA molecule (Supplementary Fig. 4). Thus, the sequence analysis of LSH10 indicates that this protein binds DNA, consistent with its proposed activity as a transcription factor.

**LSH10 is a transcriptional repressor of the OTLD1 target genes**. The interaction of LSH10 with OTLD1 and its potential DNA binding ability suggest that LSH10 may function as a transcription factor that directs the OTLD1 co-repressor to its target genes. In this scenario, LSH10 should function in complex with OTLD1 and, thus, repress at least a subset of the target genes repressed by OTLD1. Our previous study indicated that OTLD1 is involved in the transcriptional repression of five genes *OSR2*, *WUS*, *ABI5*, *ARL*, and *GA20OX* via deubiquitylation of H2B in their promoter chromatin[19]. Thus, we examined the effect of *LSH10* loss-of-function mutations on the transcription of these genes. To this end, two *Arabidopsis lsh10* T-DNA insertion lines (SALK_006965 and SK14678) were obtained from ABRC (www.arabidopsis.org/abrc/) and the homozygous mutant lines, designated *lsh10-1* and *lsh10-2*, were generated. The *lsh10-1* and *lsh10-2* mutants contained a single

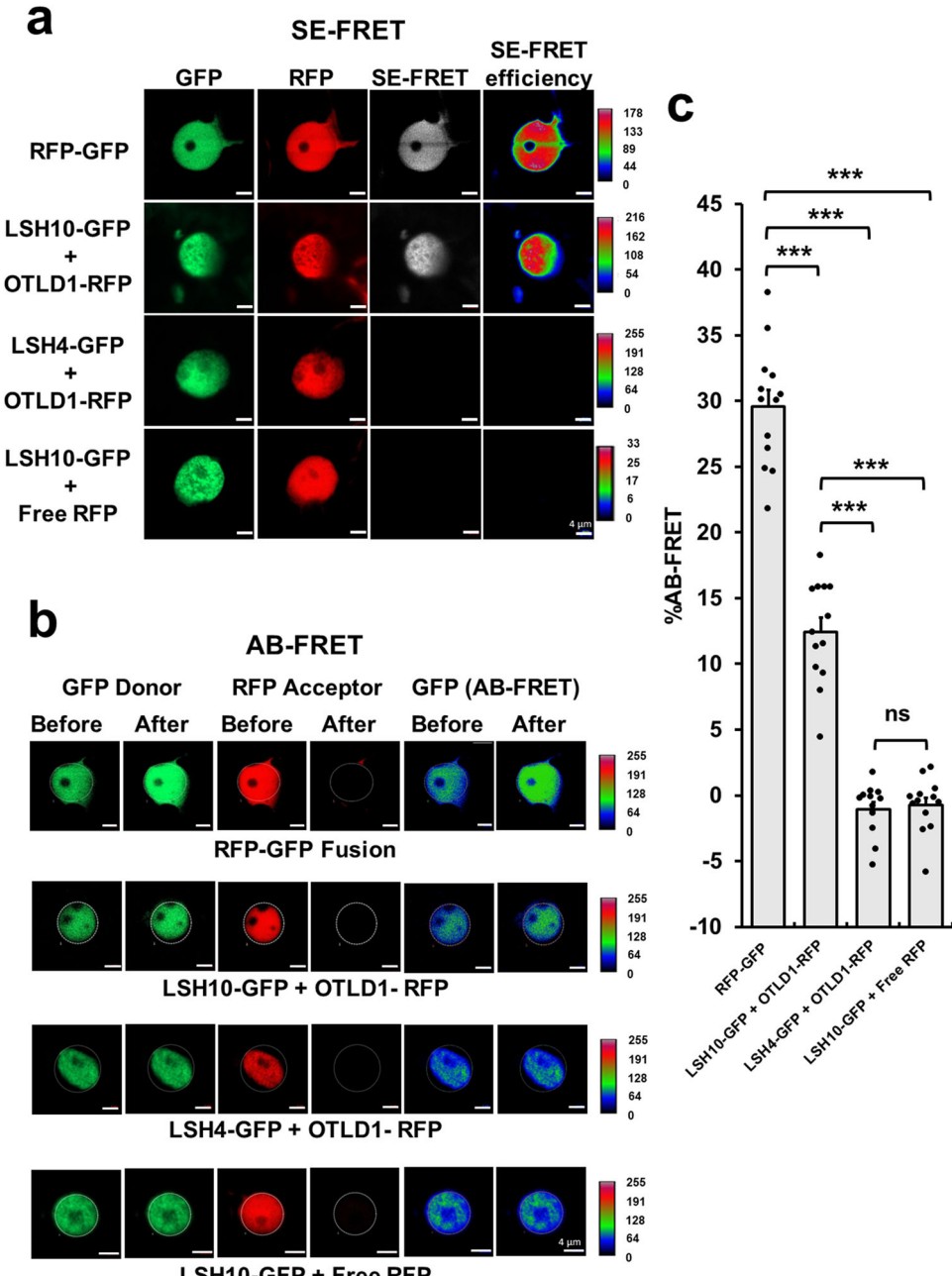

**Fig. 2 Specific interaction of LSH10 with OTLD1 *in planta* detected by FRET.** The indicated combinations of proteins tagged with GFP (energy donor) and RFP (energy acceptor) were transiently co-expressed in *N. benthamiana* leaves. **a** SE-FRET. Images were collected from the three detection channels, i.e., donor (GFP), acceptor (RFP), and raw SE-FRET. The SE-FRET efficiency images represent the calculated corrected image after subtraction of spectral bleed-through and are presented in a rainbow pseudo-color scale, in which red denotes the highest SE-FRET signal and blue denotes the lowest signal. Scale bars = 4 μm. **b** AB-FRET. Images were collected from the two detection channels, i.e., donor (GFP) and acceptor (RFP), before and after photobleaching. The circle delineates a region that was photobleached and its fluorescence measured. The AB-FRET is presented in a rainbow pseudo-color scale, in which red denotes the highest GFP signal and blue denotes the lowest signal. Scale bars = 4 μm. **c** Quantification of AB-FRET. Data measured in the experiments described in panel **b** were used to calculate %AB-FRET. Error bars represent SEM of $n = 13$ independent cells for each measurement. The individual data points are indicated, and their numerical values are listed in Supplementary Data 2. Differences between mean values assessed by the two-tailed *t*-test are statistically significant for the *p*-values *$p < 0.05$, **$p < 0.01$, and ***$p < 0.001$; $P \geq 0.05$ are not statistically significant (ns).

T-DNA insertion in the exon and 5′UTR of the *LSH10* gene, respectively (Fig. 4a). The RT-qPCR analysis showed that, in both *lsh10-1* and *lsh10-2* plants, the transcription of the *LSH10* gene was virtually abolished (Fig. 4b), confirming the loss of function of this gene in both mutant lines.

Next, the amounts of transcripts of each of the *OSR2, WUS, ABI5, ARL*, and *GA20OX* genes were analyzed by RT-qPCR in the

*lsh0-1* and *lsh10-2* plants and compared to the wild-type plants. Each of the *OSR2, WUS, ABI5*, and *ARL* genes displayed a substantial and statistically significant increase in expression in both loss-of-function lines (Fig. 4c). Specifically, the amounts of the *OSR2, WUS, ABI5*, and *ARL* transcripts were elevated ca. 24.42 to 25.43-fold, 26.99 to 24.05-fold, 20.42 to 8.45-fold and 48.23 to 66.22-fold in *lsh10-1* and *lsh10-2*, respectively. The

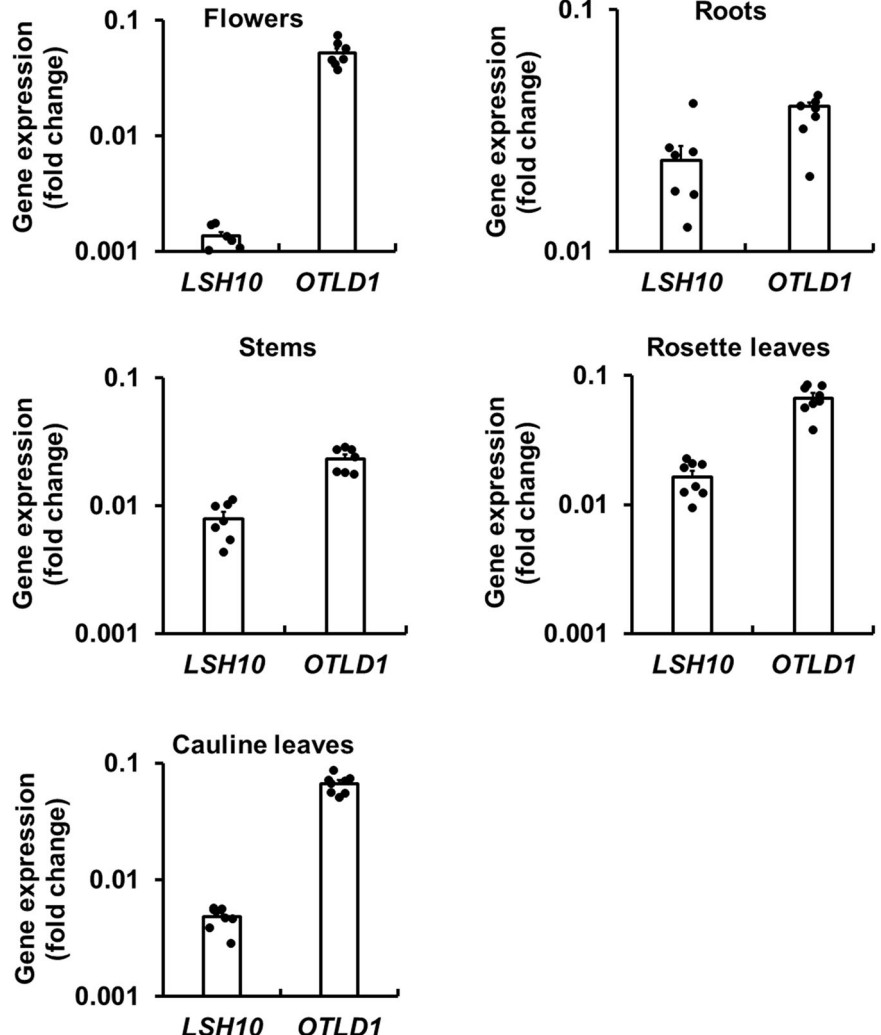

**Fig. 3 Coexpression of the endogenous *LSH10* and *OTLD1* genes in different organs of the wild-type Arabidopsis plants.** The indicated tissues were harvested from 36 days-old plants. Expression of *LSH10* and *OTLD1* was analyzed by RT-qPCR with primers listed in Supplementary Data 1. Gene expression is shown on a log scale. The individual data points are indicated, and their numerical values are listed in Supplementary Data 3. Error bars represent SEM of $n = 7$ or $n = 8$ independent biological replicates.

expression of the internal reference gene *EF1a* was not significantly altered in any of the plant lines (Fig. 4c).

Besides testing two different alleles of the *lsh10* loss-of-function mutant, we confirmed that derepression of the OTLD1 target genes resulted from the decrease in *lsh10* transcription by genetic complementation of one of the alleles, *lsh10-1*, with the wild-type *LSH10* coding sequence. We generated a transgenic *lsh10-1* line, *lsh10-1/LSH10-His6*, that expresses wild-type LSH10 protein tagged with hexahistidine. The resulting *lsh10-1/LSH10-His6* plants expressed the tagged *LSH10* at higher levels than the parental *lsh10-1* plants (compare Fig. 4d to Fig. 4b); these levels were comparable and even slightly, ca. 1.5-fold, higher than the levels of the endogenous *LSH10* transcript in the wild-type plants (Fig. 4d). In these genetically complemented plants, we observed clear repression of all four target genes, i.e., *OSR2, WUS, ABI5*, and *ARL*, relative to the loss-of-function *lsh10-1* parental plants (compare Fig. 4d to Fig. 4c) whereas no such repression was detected with the negative control *EF1a* gene (Fig. 4d). This analysis confirms that *LSH10-His6* can functionally complement the *lsh10-1* mutation. Collectively, the data in Fig. 4 indicate that LSH10 acts as a transcriptional repressor of most of the known OTLD1 target genes.

**LSH10 binds to the promoter DNA sequences and associates with the chromatin of the OTLD1/LSH10 target genes to deubiquitylate H2B.** The proposed function of LSH10 as a transcriptional repressor that facilitates the function of the OTLD1 co-repressor at the target chromatin implies that LSH10 binds directly to the regulatory sequences of the gene regulated both by OTLD1 and LSH10. Thus, we examined whether LSH10 can bind the promoters of the *OSR2, WUS, ABI5*, and *ARL* genes directly, using the electrophoretic mobility shift assay (EMSA). We selected 2–4 conserved motifs of the intergenic regions of each of these genes (Fig. 5a) and used them as EMSA probes for interaction with a purified recombinant LSH10 tagged with GST (glutathione-S-transferase). Figure 5b shows that each of these probes was recognized by LSH10 as detected by substantially reduced electrophoretic mobility of the GST-LSH10-probe complexes as compared to the free probe (lanes 3, 7, 11, 15, 19, 23, 27, 31, 35, 39 and lanes 1, 5, 9, 13, 17, 21, 25, 29, 33, 37 respectively). This binding was specific because it was not observed with GST alone (Fig. 5b, lanes 2, 6, 10, 14, 18, 22, 26, 30, 34, 38) and substantially reduced in the presence of competing amounts of unlabeled DNA, corresponding to each probe (Fig. 5b, lanes 4, 8, 12, 16, 20, 24, 28, 32, 36). Consistent with this binding specificity,

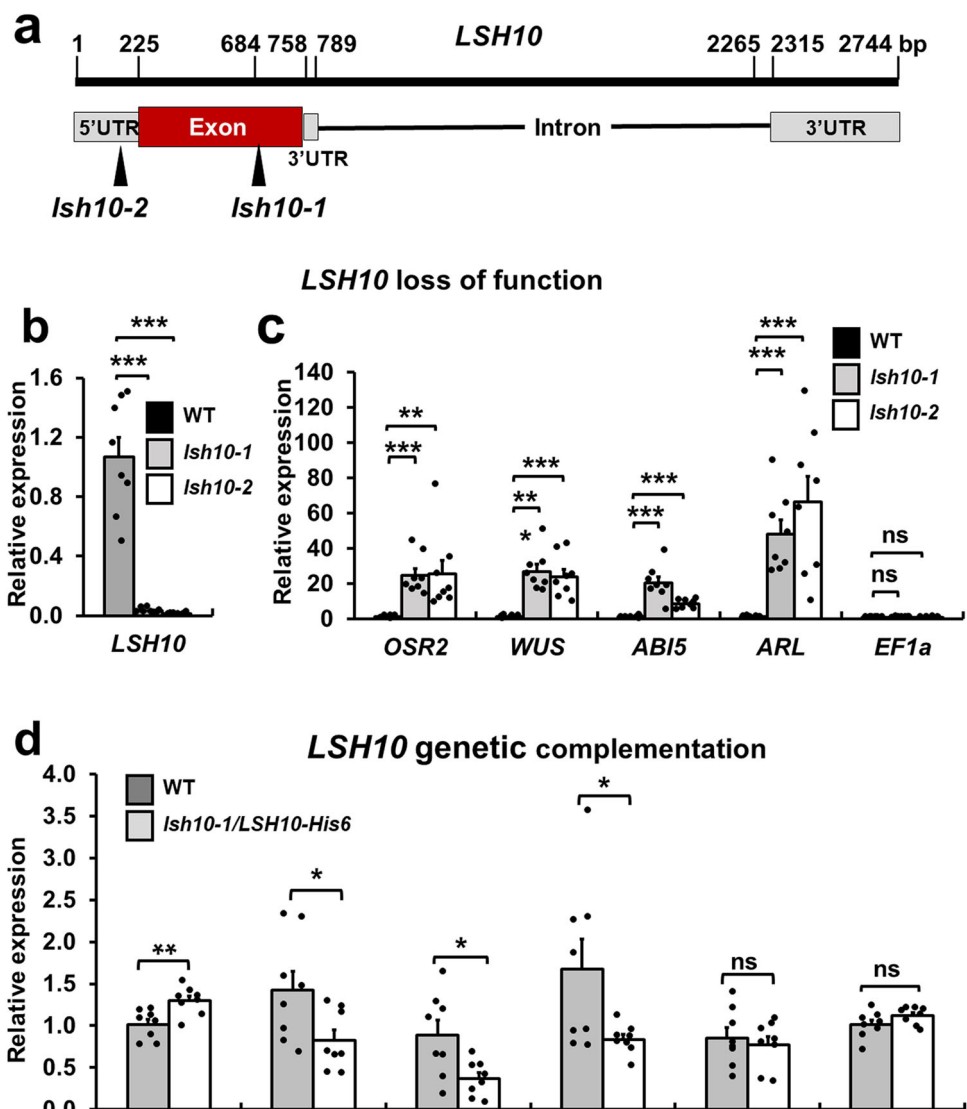

**Fig. 4 LSH10 is a transcriptional repressor of the OTLD1 target genes *OSR2*, *WUS*, *ABI5*, and *ARL*. a** A schematic structure of the *LSH10* gene and its loss-of-function alleles *lsh10-1* and *lsh10-2*. The locations of the mutagenic T-DNA inserts in each of the alleles are indicated by arrowheads. **b** Loss of the *LSH10* expression in the *lsh10-1* and *lsh10-2* plants. Wild-type plants, black bars; *lsh10-1*, gray bars; *lsh10-2*, white bars. **c** Increase in expression of the indicated target genes in the *lsh10-1*, and *lsh10-2* mutants. Wild-type plants, black bars; *lsh10-1*, gray bars; *lsh10-2*, white bars. **d** Transcriptional repression of the indicated target genes in genetically complemented *lsh10-1/LSH10-His6* plants. Wild-type plants, gray bars; *lsh10-1/LSH10-His6*, white bars. The increase in gene expression and transcriptional repression were analyzed by RT-qPCR with primers listed in Supplementary Data 1. Error bars represent SEM of $n = 8$ biological replicates. The individual data points are indicated, and their numerical values are listed in Supplementary Data 4 and Supplementary Data 5. Differences between mean values assessed by the two-tailed *t*-test are statistically significant for the *p*-values *$p < 0.05$, **$p < 0.01$, and ***$p < 0.001$; $P \geq 0.05$ are not statistically significant (ns).

not all selected motifs were recognized by LSH10 (e.g., Fig. 5b, lanes 40, 41, 42). Taken together, the EMSA experiments lend support to the idea that LSH10 functions as a DNA-binding protein that recognizes sequence elements within the target gene promoters. That several diverse promoters are recognized by LSH10 suggests a fuzzy type of recognition that allows a single transcription factor to bind variable consensus DNA sequences[38].

Next, we investigated the potential association of LSH10 with its target gene chromatin within plant cells, taking advantage of the fact that, in the *lsh10-1/LSH10-His6* line, LSH10 is tagged with an His6 epitope, allowing us to utilize quantitative chromatin immunoprecipitation (qChIP) to detect its presence. To correlate the physical association of LSH10 with the target chromatin and the binding of LSH10 to the target promoter sequence, the qChIP

primers were designed to overlap several EMSA probes (Fig. 5a). Figure 6a shows that, indeed, LSH10 was associated with the regions of the *OSR2*, *WUS*, *ABI5*, and *ARL* chromatin that contained the DNA sequences to which LSH10 was able to bind (Fig. 5). The amounts of immunoprecipitated LSH10-His6, relatively to the background signal obtained with the wild-type plants that do not express the His6 tag, were statistically significant yet varied between different target genes, potentially reflecting the different amounts of the LSH10/OTLD1-containing repressor complexes involved in the repression of each of these genes. For negative control experiments, we decided to use the *ABI5* gene, but with qChIP primers outside and upstream of the EMSA probe locations (Fig. 5a), expecting a reduced or no LSH10 association with this chromatin region. Indeed, only background-

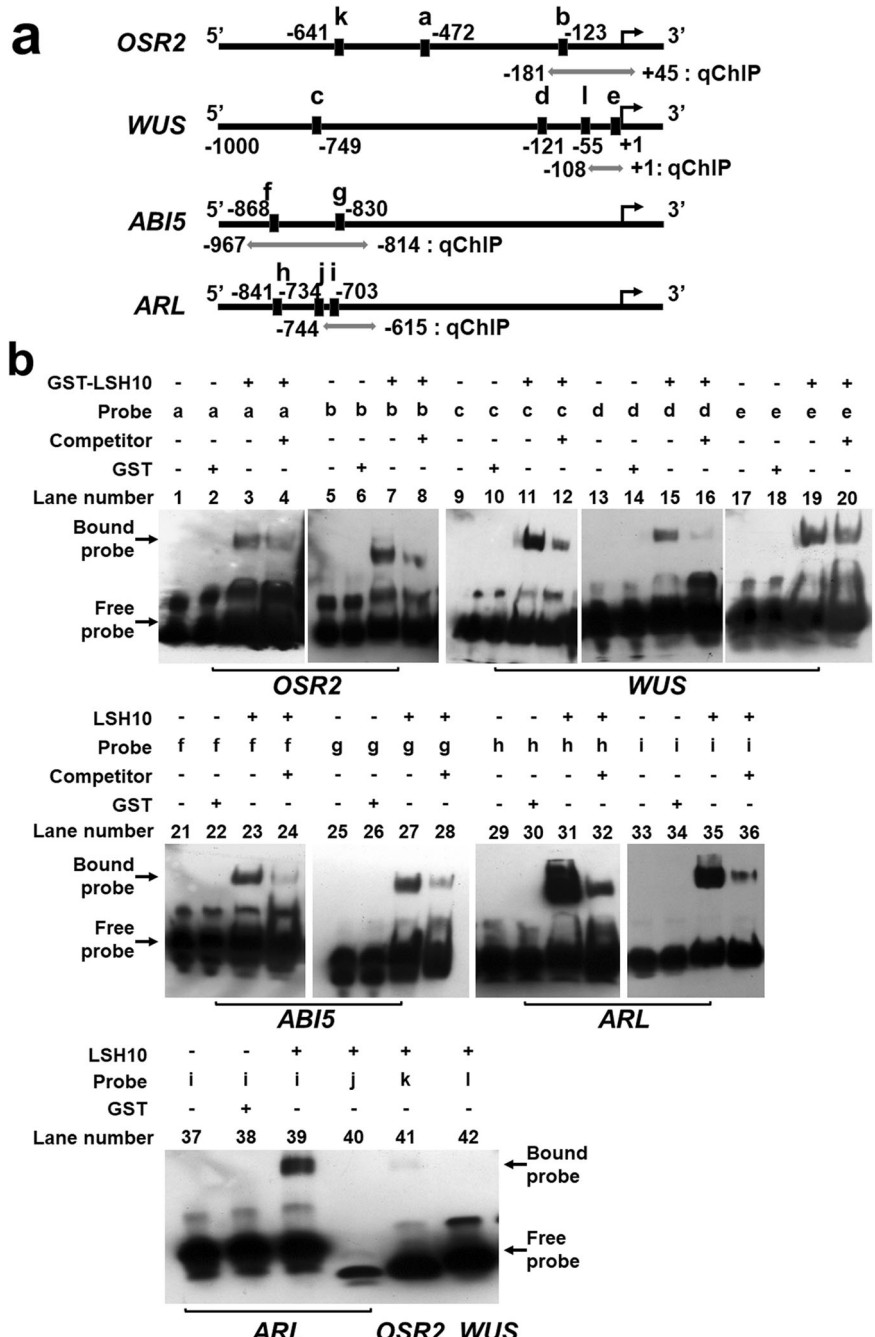

**Fig. 5 Binding of LSH10 to the promoter DNA sequences of the *OSR2, WUS, ABI5*, and *ARL* target genes. a** A schematic of the *OSR2, WUS, ABI5*, and *ARL* gene promoters showing the locations of probe sequences for EMSA (identified by black rectangles and letters "a"-"i" and detailed in Supplementary Data 1) and probes for qCHIP (denoted by double-headed gray or white arrows for probes overlapping or outside of the EMSA probe locations, respectively, and detailed in Supplementary Data 1) relative to the translation initiation sites (bent arrows). **b** EMSA of LSH10 binding to promoter sequences of the indicated target genes. The composition of the binding mixtures is detailed above each lane of the gel image, with (+) or (−) signifying the presence or absence, respectively, of the indicated components. The electrophoretic mobilities of the LSH10-bound and free probes are indicated by arrows.

level qCHIP signal was observed with these primers (Fig. 6a). Overall, these data demonstrate the ability of LSH10 to bind the conserved DNA motifs in the target gene promoters and to associate with the promoter chromatin of these genes.

Finally, we examined the notion that, if LSH10 cooperates with OTLD1 to deubiquitylate the target chromatin, such deubiquitylation will be reduced in the absence of LSH10. Thus, we used qCHIP to analyze the promoter chromatin of the *OSR2, WUS, ABI5*, and *ARL* genes in the loss-of-function *lsh10-1* and

*lsh10-2* mutants in and in the wild-type plants for the presence of monoubiquitylated H2B, known to be deubiquitylated by OTLD1 in these chromatin regions[27]. Figure 6b shows a statistically significant degree of hyperubiquitylation of H2B in the *OSR2, WUS, ABI5*, and *ARL* chromatin in both mutants as compared to the wild-type plants. Specifically, the *OSR2* chromatin of *lsh10-1* and *lsh10-2* plants was monoubiquitylated on average 4.24-2.46-fold more than the wild-type *OSR2* chromatin, and monoubiquitylation of *WUS, ABI5*, and *ARL*

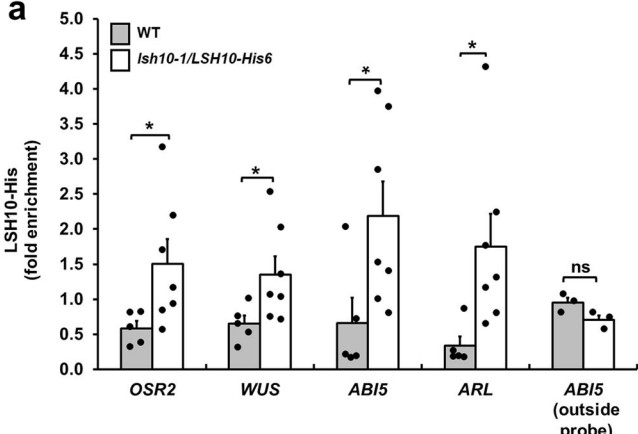

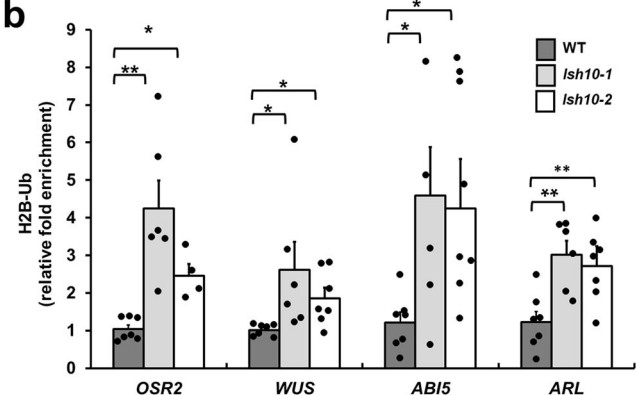

**Fig. 6 Association of LSH10 with and effects on ubiquitylation of the chromatin of the *OSR2, WUS, ABI5*, and *ARL* genes. a** Association of LSH10-His6 with the chromatin of the indicated target genes in the *lsh10-1/LSH10-His6* plants. *lsh10-1/LSH10-His6*, white bars; non-specific background signal obtained in the wild-type plants that do not express the His6 epitope, gray bars. The outside probe is the qChIP probe, depicted in Fig. 5a, and located outside the EMSA probe locations. **b** Increase in H2B monoubiquitylation of the *OSR2, WUS, ABI5*, and *ARL* promoter chromatin in the *lsh10-1* mutant plants. Wild-type plants, dark gray bars; *lsh10-1*, light gray bars; *lsh10-2*, white bars. The chromatin association of LSH10-His6 and the degree of H2B monoubiquitylation were analyzed by qChIP with probes described in Fig. 5a and listed in Supplementary Data 1. Error bars represent the SEM of $n = 5$ or $n = 7$ biological replicates with 3 technical repeats for each. The individual data points are indicated, and their numerical values are listed in Supplementary Data 6 and Supplementary Data 7. Differences between mean values assessed by the two-tailed t-test are statistically significant for the p-values *$p < 0.05$ and **$p < 0.01$; $P \geq 0.05$ are not statistically significant (ns).

was increased 2.62-1.86-fold, 4.59-4.25-fold, and 3.02-2.71-fold, respectively (Fig. 6b).

## Discussion

Epigenetic regulation of transcriptional outcomes heavily relies on the action of histone-modifying enzymes that function as writers and erasers of euchromatic and heterochromatic marks. Most of these enzymes, however, lack DNA binding capabilities and, therefore, are unable to recognize their target chromatin. Instead, this targeting most likely is mediated by DNA-binding transcription factor proteins that recognize specific histone-modifying enzymes and recruit them to their sites of function. For example, the transcription factor SUF4 (SUPPRESSOR OF FRIGIDA4) likely recruits the histone methyltransferase EFS

(EARLY FLOWERING IN SHORT DAYS) and the PAF1-like complex to the floral inhibitor *FLC* (FLOWERING LOCUS C) promoter[39,40]. The transcription factor ALF (ABI3-like factor) is thought to recruit a histone acetyltransferase (HAT) activity to the *phaseolin* promoter in the bean (*Phaseolus vulgaris*)[41]. Transcription factors BPC1 (BASIC PENTACYSTEINE 1), BPC6, and AZF1 (ARABIDOPSIS ZINC FINGER 1) interact with the components of Polycomb repressive complexes PRC2 and PRC1 and recruit them to the GAGA and telobox motifs to promote histone 3 (H3) trimethylation and H2A monoubiquitylation[42,43]. Transcription factors VAL1 (VP1/ABI3-like 1) and VAL2 interact with the PRC1 component LHP1 and recruit it to the BY motifs[44] whereas NAC (NAM, ATAF, CUC) transcription factors NAC050 and NAC052 interact with and recruit the histone demethylase JMJ14 to its target genes[45]. This relatively short list of plant histone-modifying enzymes that are recognized and recruited by transcription factors does not include histone deu-biquitinases, one of the main histone modifiers for which the mechanism of recruitment is poorly understood.

Here, we began filling this gap in our knowledge by detecting the specific interaction between the OTLD1 histone deubiquiti-nase and the LSH10 protein of Arabidopsis. LSH10 belongs to the ALOG family of proteins, such as DUF640 (domain of unknown function 640), known to act as key developmental regulators in land plants. The presence of DNA binding motifs, transcriptional regulation activity, nuclear localization, and homodimerization ability[32,46–48] suggest that the ALOG proteins may function as specific transcription factors. Our present understanding of the biological functions of different members of the ALOG protein family is very limited. Several *ALOG/LSH* (*LIGHT-SENSITIVE SHORT HYPOCOTYLS*) genes have been characterized in Ara-bidopsis. The initially identified Arabidopsis *LSH* gene is *LSH1*, the protein product of which is involved in the light-dependent regulation of hypocotyl development[49]. Subsequently, LSH3 and LSH4 were shown to suppress the differentiation of organs, such as cotyledons, leaves, and flowers[47,50,51], LSH8 was reported to have a role in the regulation of the ABA signaling pathway[52], and LSH9 was shown to regulate hypocotyl elongation by interacting with the temperature sensor ELF3[53,54]. The function of LSH10, however, remained unknown, and our observations here suggest that it can function as a transcriptional repressor that interacts with the co-repressor histone deubiquitinase OTLD1 for its functional recruitment to promoters of several of its target genes.

The LSH10-OTLD1 interaction was initially detected in yeast and then confirmed by two independent approaches, BiFC and FRET, within living plant cells. In addition to demonstrating the interaction *in planta*, BiFC established the subcellular localization of the LSH10-OTLD1 complexes in the nucleus whereas FRET indicated that, in this complex, the interacting proteins are positioned <10 nm from each other. The likely function of LSH10 in these complexes was investigated using LSH10 loss-of-function and gain-of-function plant lines, both of which indicated that LSH10 acts as a transcriptional repressor and that its target genes include four of the five genes known to be repressed by OTLD1[19]. Specifically, LSH10 repressed the *OSR2, WUS, ABI5*, and *ARL* genes whereas the transcription of the *GA20OX* gene was not affected by LSH10. Consistent with this activity, LSH10 is asso-ciated with the chromatin of the promoter regions of each of the *OSR2, WUS, ABI5*, and *ARL* genes in plant cells. In the LSH10 loss-of-function plants, these promoter regions were enriched for monoubiquitylated H2B, suggesting impairment in the OTLD1 recruitment to the target chromatin. Furthermore, LSH10 directly associated with the DNA sequences of these promoter regions. The DNA binding activity of LSH10 detected experimentally is supported by the alignment of the secondary structure of LSH10 predicted by AlphaFold[55,56] (Supplementary Fig. 2a) with the

known DNA binding domain of the Cre recombinase (PDB: 1ouqA) (Supplementary Fig. 2b); this alignment, using the TM-align algorithm[57], suggests that both structures are in a similar fold with the TM-score of 0.59682, with the LSH10 helices 1–4 merged with the Cre recombinase DNA binding domain helices B-E, respectively (Supplementary Fig. 2c).

In summary, our observations suggest that LSH10 functions as a specific transcription factor that recognizes OTLD1 and mediates its functional recruitment to the target gene promoters, and that OTLD1 may utilize multiple transcription factors for this purpose, depending on the specific target gene. The latter notion of several, and most likely at least partly redundant pathways for transcriptional control of the OTLD1 target genes may explain why both *LSH10* loss-of-function mutant alleles did not produce visually discernable morphological or developmental phenotypes. Interestingly, OTLD1—whose loss-of-function mutants also have no detectable phenotypes[19]—has been suggested to share functional redundancy with some of its 12 OTU family homologs[16,19,58,59]. Note that our study did not endeavor to understand the specific phenotypic effects of LSH10; instead, we aimed to uncover the role of LSH10 as a transcription factor that recognizes and directs OTLD1 to the target gene promoters as a paradigm for the mechanism by which plant co-repressor histone deubiquitinases are recruited to the target chromatin by their cognate DNA binding repressors. The limation is that this study only focuses on the functional interaction between the Arabidopsis transcription factor LSH10 and histone deubiquitinase OTLD1 during transcriptional regulation of a common subset of their target genes. The biological details of this interaction should be elucidated by producing Arabidopsis transgenic lines, in which LSH10 and OTLD1 tagged with different autofluorescent reporters will be expressed from their native promoters and used to demonstrate physical existence of both proteins in the same cell, consistent with their functional cooperation in regulation of transcription. Then, another subset of transgenic plants should be generated, in which epitope-tagged OTLD1 will be expressed in the LSH10 loss-of-function mutants, lsh10-1 and/or lsh10-2; these lines will be useful for demonstration of the physical absence of OTLD1 from the target chromatin in the absence of LSH10.

## Methods

**Plants and growth conditions**. The *Arabidopsis thaliana* loss-of-function T-DNA insertional mutants of *LSH10* (*lsh10-1*, SALK_006965; *lsh10-2*, SK14678) were obtained from ABRC (www.arabidopsis.org/abrc/). For genetic complementation, gain-of-function lines of the lsh10-1 mutant, the Arabidopsis *LSH10* cDNA was amplified using primers listed in Supplementary Data 1, cloned into pDONR207 (#12213013, Invitrogen) by the BP reaction using the Gateway BP Clonase II (#11789100, Invitrogen), and transferred into the destination vector pMDC32[60] by the LR reaction using the Gateway LR Clonase II (#11791020, Invitrogen). The recombinant plasmids were subsequently introduced into *Agrobacterium tumefaciens* strain GV3101, and transformed into the *lsh10-1* mutant plants by the floral dip method[61]. The transgenic plants were selected on MS medium supplemented with hygromycin (30 mg/l) and timentin (100 mg/l) and confirmed by PCR and RT-qPCR using primers listed in Supplementary Data 1.

*Nicotiana benthamiana* plants, wild-type *Arabidopsis thaliana* (ecotype Col-0) plants, and the mutant lines were grown on soil in an environment-controlled chamber at 22 °C under long-day conditions (16-h light/8-h dark cycle at 140 µE sec$^{-1}$m$^{-2}$ light intensity) as described previously[18,19]. The use of these plant species is helpful because the consistent findings in different systems bolster the robustness of the reported data.

**Bimolecular fluorescence complementation (BiFC) assay and subcellular localization assay**. Arabidopsis *OTLD1, LSH10, LSH4* cDNAs, and the *Tobacco mosaic virus* methyltransferase domain (MT) coding sequence were amplified using primers listed in Supplementary Data 1, cloned into pDONR207, and transferred into the destination vectors pGTQL1221YC (#61705, Addgene) and pGTQL1211YN (#61704, Addgene). For BiFC, different combinations of the resulting constructs encoding the nEYFP- and cEYFP-tagged proteins were transiently co-expressed with free CFP in the leaves of 4–8-week-old *N. benthamiana*

plants by agroinfiltration. Free CFP was expressed from pPZP-RCS2A-DEST-ECFP-C1[62]. For subcellular localization, LSH10 was fused with CFP by transferring its coding sequence by Gateway cloning from pDONR207 into pPZP-RCS2A-DEST-ECFP-N1[62]. The fluorescence signal was recorded 2 days after infiltration using a confocal laser scanning microscope (LSM 900, Zeiss, Germany) with a 40X oil immersion objective and CFP and YFP filters.

**Fluorescence resonance energy transfer (FRET) assay**. The coding sequences of *LSH10* and *OTLD1* were transferred by Gateway cloning from pDONR207 into pPZP-RCS2A-DEST-EGFP-N1[62] and pPZP-RCS2A-DEST-ERFP-N1 to generate the donor vector p35S::LSH10-GFP and the acceptor vector p35S::OTLD1-RFP, respectively. For positive control, ERFP was amplified from pPZP-RCS2A-DEST-ERFP-N1 using primers listed in Supplementary Data 1, cloned into pDONR207, and transferred into pPZP-RCS2A-DEST-EGFP-N1 to generate the RFP-GFP fusion construct. These vectors were transiently expressed in the leaves of 4–8-week-old *N. benthamiana* plants via agroinfiltration. The FRET signal was detected and recorded by confocal microscopy using a 40X oil immersion objective.

For sensitized emission, a set of three confocal images of the same field of view was taken using the following channel settings: the GFP channel for excitation and emission of the donor chromophore (excitation lasers: 405 nm, emission filter: 400–597 nm), the RFP channel for excitation and emission of the acceptor chromophore (excitation lasers: 561 nm, emission filter: 400–597 nm), and the FRET channel for excitation of the donor and emission of the acceptor chromophores (excitation lasers: 405 nm, emission filter: 597–617 nm); the RFP channel emission filter for was set to separate the acceptor signal from the FRET signal. The ImageJ plug-in PixFRET software was used to generate corrected images of SE-FRET efficiency after subtraction of spectral bleed-through. The bleed-through values for the donor and the acceptor were obtained from images of cells expressing each of the constructs alone (Supplementary Fig. 5).

For acceptor photobleaching, the emission of the donor fluorophore is compared before and after the photobleaching of the acceptor. Photobleaching of the acceptor leads to an increase in the donor fluorescence if any interactions leading to energy transfer occur because it is no longer quenched by the acceptor. Acceptor photobleaching was performed with 100% intensity of lasers at 561 nm, duration of 30 s, 150 interactions for area bleach, and started after 5 images. Images in the acceptor channel (RFP) and donor channel (GFP) were captured simultaneously before and after photobleaching. %AB-FRET was calculated as the percent increase in GFP emission after RFP photobleaching using the following Eq. (1):

$$\%AB-FRET = ((GFPpost - GFPpre) \div GFPpre) \times 100 \qquad (1)$$

where GFPpost is GFP emission after RFP photobleaching, and GFPpre is GFP emission before RFP photobleaching. The %AB-FRET was determined in regions of interest drawn around the entire area of the cell nucleus.

**DNA, RNA extraction, and reverse transcription-quantitative PCR (RT-qPCR)**. Plant genomic DNA was extracted using the DNeasy-like Plant DNA Extraction Protocol. RNA from plants was extracted using TRIzol (#15596026, Invitrogen) according to the manufacturer's instructions. The RNA was used as a template for cDNA synthesis *via* the RevertAid Reverse Transcription Kit (#K1691, Thermofisher) and Hexa-random primers. RT-qPCR was performed using Power SYBR Green PCR master mix (Catalog #: 4367659) (Applied Biosystems by Thermo Fisher Scientific, USA) and a QuantStudio™ 3 Real-Time PCR System (Applied Biosystems by Thermo Fisher Scientific). The thermocycler program consisted of pre-denaturation at 95 °C for 10 min followed by 40 cycles at 95 °C for 15 s and 60 °C for 1 min. Each sample was analyzed in 8 biological replicates and 3 technical repeats for each, and the relative expression values were calculated using the $2^{-\Delta\Delta Ct}$ formula[63], where Ct indicates cycle threshold, i.e., the number of PCR cycles required for the signal to become detectable above the background. *A. thaliana Actin, Sand*, and *EF1a* were used as reference genes. An averaged Ct value from all three of these genes was also used for the normalization of data in the calculation of fold changes in expression of the endogenous *LSH10* and *OTLD1* genes in different organs of the wild-type plants using the ΔCt-method as described[64]. Specific primers used in these experiments are detailed in Supplementary Data 1.

**qChIP and statistical analysis**. Approximately 0.8–1.0 g of leaves of 4-week-old *lsh10-1/LSH10-His6* and wild-type plants were harvested and cross-linked with 1% formaldehyde (v/v) for quantitative chromatin immunoprecipitation (qChIP) assays. The qChIP experiments were performed using an EpiQuikTM Plant ChIP Kit (EpiGentek, P-2014-48). According to the manufacturer's instructions, the chromatin from the plant cells was extracted and sheared; the length of sheared DNA fragments was 200–1000 bp. 5 µl of the crude chromatin extracts were saved for use as an input control, and 100 µl were added into the microwell immobilized with the antibodies (non-specific rabbit IgG Isotype Control (Invitrogen, WB317638, 1 µg) as the negative control, specific rabbit anti-His-tag antibody (GenScript, A00174-40, 2.5 µg) or anti-monoubiquityl-histone H2B (Lys-120) (5546S, Cell Signaling Technology, Inc., 2.5 µg). DNA was released from the antibody-captured protein-DNA complex, reversed, purified through the

specifically designed F-Spin Column, and then amplified by qPCR as described above. Input Ct values were adjusted for the dilution factor and ΔCt was calculated by normalizing Ct values to the adjusted input based on the Eq. (2)

$$\Delta Ct = Ct(input) - Ct(IP) \qquad (2)$$

For anti-His6 qChIP, the % input was calculated as Eq. (3)

$$\%input = 100 \times 2e - \Delta Ct \qquad (3)$$

Fold enrichment was calculated using Eq. (4)

$$\%input \ of \ \frac{IP}{IgG} = \Delta Ct(IP) \div \Delta Ct(IgG) \qquad (4)$$

For anti-H2Bub qChIP, the relative fold enrichment was calculated using the $2^{-\Delta\Delta Ct}$ formula and normalized with adjusted input[65].

**Electrophoretic mobility shift assays (EMSAs).** The full-length cDNA of *LSH10* was cloned into the GST fusion vector pGEX-5X-1 and propagated in the LEMO21 strain of *Escherichia coli*. The recombinant protein GST-LSH10 was purified using Glutathione Sepharose 4B beads (GE Healthcare, 17-0756-01) according to the manufacturer's protocol. The probes were designed by two iterations of motif-based sequence analysis of the intergenic regions of the *ARL*, *WUS*, *ABI5*, and *OSR2* genes for conserved motifs using the Multiple Expectation Maximizations for Motif Elicitation (MEME) tool (https://meme-suite.org/meme/). The biotin-labeled and unlabeled oligonucleotides corresponding to both strands of the selected sequences (see Fig. 5a and Supplementary Data 1) were synthesized by Integrated DNA Technologies (Coralville, IA). The oligonucleotides were annealed, and the resulting probes (10 ng) were incubated with the purified protein (1 μg) and dI.dC (1 μg) at room temperature for 20 min in the binding buffer (100 mM Tris, 500 mM KCL, 10 mM DTT, pH 7.5). For competition experiments, a 300-fold molar excess of each unlabeled probe was included in the binding reaction. EMSA was performed using the Light Shift Chemiluminescent EMSA kit (Thermo Scientific) according to the manufacturer's instructions. The electrophoretic migration of biotin-labeled probes was resolved on 6% native polyacrylamide gels and detected using an enhanced chemiluminescence substrate (Thermo Scientific), followed by autoradiography. The uncropped images of the EMSA gels are shown in Supplementary Fig. 6.

**Statistics and reproducibility.** Statistical significance was determined by paired two-tailed Student's t-test using Excel Microsoft 365 software. *P* values are indicated as $*p < 0.05$, $**p < 0.01$, and $***p < 0.001$ corresponding to the statistical probability of >95%, 99%, and 99.9%, respectively, considered statistically significant. ns is indicated not significant with $p \geq 0.05$. Quantitative data are shown as the means of at least five biological replicates ($n > 5$). Error bars represent SEM of independent biological replicates.

**Reporting summary.** Further information on research design is available in the Nature Research Reporting Summary linked to this article.

## Data availability
Primers and Probes information using in this study are shown in Supplementary Data 1. Raw data for quantification of AB-FRET (for Fig. 2c) is provided in Supplementary Data 2. Raw data for quantification of the co-expression of the LSH10 and OTLD1 genes in different organs of the wild-type Arabidopsis plants (for Fig. 3) is provided in Supplementary Data 3. Raw data for quantification of the increase in expression of the target genes in the lsh10-1, and lsh10-2 plants (for Fig. 4b, c) is provided in Supplementary Data 4. Raw data for quantification of the transcriptional repression of the target genes in the lsh10-1/LSH10-His6 plants (for Fig. 4d) is provided in Supplementary Data 5. Raw data for quantification of the qChIP analysis of the association of LSH10-His6 with the chromatin of the target genes (for Fig. 6a) is provided in Supplementary Data 6. Raw data for quantification of the qChIP analysis of the increase in H2B monoubiquitylation of the target chromatin (for Fig. 6b) is provided in Supplementary Data 7. Uncropped EMSA gel images are shown in Supplementary Figs. 6–8.

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

## Acknowledgements

The work in the V.C. laboratory was supported by grants from NIH (R35GM144059 and R01GM50224), NSF (MCB1913165 and IOS1758046), and BARD (IS-5276-20) to V.C. and M.L.

## Author contributions

M.S.V.P. conducted the experiments and analyzed the experimental data. P.T.T. conducted FRET. I.K. initiated the identification of the *LSH10* gene. M.S.V.P. and V.C. designed the experiments and, with the help of M.L., wrote, reviewed, and edited the manuscript.

## Competing interests

The authors declare no competing interests.
