## [Peer Review File · Communications Biology]

Reviewers' comments:

Reviewer #1 (Remarks to the Author):

This well-written and interesting manuscript by Phan et al. details their discovery of the likely mechanism of recruitment of a plant histone deubiquitinase OTLD1 to its target chromatin by a specific transcription factor LSH10. The interaction between OTLD1 and LSH10 was convincingly demonstrated using two independent approaches, both in living plant tissues. LSH10 and OTLD1 associated with the chromatin of the same target genes and the effects of LSH10 genetic knockout were studied in two loss-of-function alleles as well as in gain-of-function transgenic plants. Technically, the experiments were soundly designed and well executed, including proper statistical analyses of the data where appropriate. The Methods section is adequate for reproducing the experiments (but see comment 5 below).

Conceptually, this paper is important. To my knowledge, so far, only one histone deubiquitinase, a mammalian BAP1, has been reported to interact with a bona fide transcription factor FOXK2 for chromatin targeting. No mechanisms for histone deubiquitinase recruitment to the target genes have been described. Thus, this study fills an important gap in our knowledge of epigenetic regulation by histone deubiquitinases, and as such will be useful to many readers in plant and non-plant research communities.

I do have several suggestions to improve the manuscript:

1. Plastid autofluorescence panels in Fig. 1 are not informative, especially since this information is shown in the merged images anyway and can be safely deleted to save space and simplify the Figure.
2. Fig. 3 could be moved to the supplementary material.
3. Fig. S1 shows a phylogenetic analysis of Arabidopsis LSH proteins. However, because this information is supplemental and, as such, does not affect the size limits of the published paper, I suggest expanding this Figure by adding another panel with a phylogenetic tree that includes LSH proteins from other plant species.
4. Antibody dilutions should be described in the relevant sections of Methods.
5. For pixel-by-pixel analyses using the ImageJ software, the generation of SE-FRET images for the protein pairs presented in Fig. 2 usually requires obtaining images of OTLD1-mRFP/free mRFP alone and LSH10-GFP/LSH4-GFP alone to determine the spectral bleed-through values for acceptors and donors, respectively. To make the SE-FRET analysis in Fig. 2 more complete and reproducible, I suggest including these images in supplementary "raw" data.
6. All bar diagrams should include data points.

Reviewer #2 (Remarks to the Author):

This manuscript provides evidence that the Arabidopsis LSH10 transcription factor interacts with the co-repressor histone deubiquitinase OTLD1. The biological relevance is that this novel type of interaction is utilized to guide OTLD1 to the target genomic sequences.

The experiments are carefully conducted and the findings are purportedly new for the type of deubiquitinases under study.

The authors use three experimental systems: *N. benthamiana*, *A. thaliana*, and *E. coli*. This is sometimes used by critics as 'argu-fodder' that findings for one system cannot be automatically

translate to others. In this case the authors c/should use their findings as an argument that the findings in different systems bolster the robustness of the reported interaction for the system because the results in three different experimental platforms point in the same direction.

Perhaps the authors together with NatCom editorial staff wish to generate a summarizing drawing/diagram that illustrates the proposed (inter)action and that could potentially be used on the cover page.

It was a pleasure to read this manuscript.

Reviewer #3 (Remarks to the Author):

The work of Mi Sa Vo Phan investigates the role of the LSH10 factor as potential recruiter of the histone deubiquitinase OTLD1.

The authors uses various technique to assess the direct protein interaction between LSH10 and OTLD1, the ability of LSH10 to bind DNA directly, and the relation between LSH10 and OTLD1 in the transcriptional regulation of a set of common target genes.

The work is logically presented, rigorously conducted, and provides insight on a interesting topic about the dynamic of histone modifications correlated with the regulation of transcription.

I have some comments regarding some experiments and conclusions drawn by that:

- Figure 4B: authors check expression levels of four OTLD1 target genes in *lsh10* mutant background, and find that WUS expression levels are higher of about 30folds in the mutant respect to the wild type. Are these plants showing any developmental defects? Overexpression of WUS can cause meristem enlargement, and also strong root phenotype. It would be very interesting to correlate the WUS levels with developmental defects. On this direction, it would be very interesting to check if the target has an expansion of the expression domain or not (as with in situ or by introgressing the marker line in the *lsh10* mutant bakcground). I believe this is the case, considering that authors have used as material for their Real-Time PCR leaves, where normally WUS is not expressed.

- Figure 4C: the complementation line *pLSH10::LSH10-6xHis* shows higher expression levels that of the endogenous LSH10 transcripts in wild-type background. Authors then correlate this increase of expression with a reduction in the expression of the target genes *OSR2*, *WUS*, *ABI5*. This analysis is not sufficient to state such direct correlation. Authors should present more transgenic lines, ideally a minimum of three independent transformants with different degrees of LSH10 expression. It is also known that an increase in mRNA levels can not correlate with an higher abundance of the protein. To make their conclusion solid, authors should couple expression level of LSH10 with analysis of protein level by Western Blot.

- Figure 6A Authors check the direct binding of LSH10 to target genes promoter. As positive binding is observed for all the gene tested, it would be nice to add also a negative control where no binding is detected. I am not familiar with the use of "A.U. - Arbitrary Unit " to present ChIP enrichments, and because of this, I find the chart incomplete and hard to understand. I apologize for this lack of expertise from my side, and I thus ask if it is possible to present the ChIP data in Figure 6A as relative fold enrichment as done for Figure 6B.

- The model that the author proposes in that LSH10 is recruiter of OTLD1. However two fundamental experiments for supporting this hypothesis are missing from the current format of the work:

- 1) co-localization of both proteins in the same cell. The work presents evidence for direct LSH10-OTDL1 protein interaction, but for this to have a biological meaning, both proteins need to be present in the same cell (nucleus) and at the same time. Therefore expression analysis in support of this is essential (either by live imaging of -GFP tagged line for instance, or by In situ hybridization).

- 2- For LSH10 to be stated as true recruiter of OTLD1, ChIP on H2B-Ub is not sufficient. Authors should assess OTLD1 binding to the target genes in *lsh10* background. If LSH10 is a true OTLD1 recruiter, than the OTLD1 binding to the target should be abolished.

Minor comments:

- The introduction would benefit with the addition of some background information on what histone ubiquitination/deubiquitination is and the roles of this modification in transcription;
- When analyzing the LSH10 protein structure, authors could add a 3D model, as the one reported here <https://www.uniprot.org/uniprotkb/Q9S7R3/entry>
- Figure 4B: expression analysis of OTLD1 targets expression in lsh10 mutant backgrounds. It would be useful to have also the expression level of such genes in the otd1 mutant, to directly compare the expression levels in the two genetic backgrounds.
- Figure 6B: ChIP of H2B-Ub in lsh10 background. It would be useful to add also here the degree of H2B-Ub enrichment in otd1 mutant background.
- Line 52: "To achieve this regulation, histone-modifying enzymes are thought to function in complexes with transcription factors that contain DNA-binding domains and, thus, can provide the DNA binding capacity to the histone modifier-transcription factor complex and thereby recruit histone-modifying enzymes to the target promoters. I find this sentence very wordy and I had to read it multiple times to understand it properly. Authors could break this into two sentences to make it clearer.
- Line 55: "Indeed, in plants, different transcription factors have been shown to recruit such diverse histone modifiers as histone methyltransferases, histone acetyltransferases, histone demethylases, and Polycomb repressive complexes that promote histone trimethylation and monoubiquitylation". Please provide some references in support of this sentence.

Reviewer 1

We thank the reviewer for the kind evaluation of our work and for important suggestions to improve the paper, all of which were addressed as follows.

1. Plastid autofluorescence panels in Fig. 1 are not informative, especially since this information is shown in the merged images anyway and can be safely deleted to save space and simplify the Figure.

As requested, the autofluorescence panels in Fig 1 were removed.

2. Fig. 3 could be moved to the supplementary material.

As requested, Fig. 3 was moved to the supplemental material and is referred to as Fig. S2 in the revised paper.

3. Fig. S1 shows a phylogenetic analysis of Arabidopsis LSH proteins. However, because this information is supplemental and, as such, does not affect the size limits of the published paper, I suggest expanding this Figure by adding another panel with a phylogenetic tree that includes LSH proteins from other plant species.

Our phylogenetic analysis of the Arabidopsis LSH proteins was extended to include other plant species and is presented in Fig. S3 of the revised paper.

4. Antibody dilutions should be described in the relevant sections of Methods.

Unlike western blotting, the qChip protocol does not dilute the antibody; instead, the manufacturer's protocol directs to use 1 ug for the IgG control and 2-3 ug for the antibody of interest. In our case, we used 1 ug of the non-specific rabbit IgG Isotype Control, 2.5 ug of the rabbit anti-His, and 2.5 ug of the anti-monoubiquityl-histone H2B. This information was added to the revised Methods.

5. For pixel-by-pixel analyses using the ImageJ software, the generation of SE-FRET images for the protein pairs presented in Fig. 2 usually requires obtaining images of OTLD1-mRFP/free mRFP alone and LSH10-GFP/LSH4-GFP alone to determine the spectral bleed-through values for acceptors and donors, respectively. To make the SE-FRET analysis in Fig. 2 more complete and reproducible, I suggest including these images in supplementary "raw" data.

The requested images were included in Fig. S5 of the revised paper.

6. All bar diagrams should include data points.

We added the data points to the graphs. We also listed the exact numerical values of all data points in the corresponding Tables S2-S7.

Reviewer 2

We thank the reviewer for the thoughtful and kind evaluation of our work.

The authors use three experimental systems: N. benthamiana, A. thaliana, and E. coli. This is sometimes used by critics as 'argu-fodder' that findings for one system cannot be automatically translate to others. In this case the authors c/should use their findings as an argument that the findings in different systems bolster the robustness of the reported interaction for the system because the results in three different experimental platforms point in the same direction.

Brilliant! we actually included the gist of this idea in the revised paper.

Perhaps the authors together with NatCom editorial staff wish to generate a summarizing drawing/diagram that illustrates the proposed (inter)action and that could potentially be used on the cover page.

Thank you for the idea. We will use our (limited) artistic abilities to design such a drawing.

It was a pleasure to read this manuscript.

It was an even larger pleasure to read this review.

Reviewer 3

We thank the reviewer for thoughtful suggestions to improve the paper, all of which were addressed as follows.

Major comments:

*Figure 4B: authors check expression levels of four OTLD1 target genes in *lsh10* mutant background, and find that *WUS* expression levels are higher of about 30folds in the mutant respect to the wild type. Are these plants showing any developmental defects? Overexpression of *WUS* can cause meristem enlargement, and also strong root phenotype. It would be very interesting to correlate the *WUS* levels with developmental defects. On this direction, it would be very interesting to check if the target has an expansion of the expression domain or not (as with *in situ* or by introgressing the marker line in the *lsh10* mutant background). I believe this is the case, considering that authors have used as material for their Real-Time PCR leaves, where normally *WUS* is not expressed.*

The reviewer is correct, and our data in Fig. 4C show only very low expression of *WUS* in the wild-type leaves; in fact, this low expression of *WUS* in leaves may be low due, at least in part, to the repressive action of the LSH10/OTLD1 repressor complex and, therefore, increase when this complex is inactivated. The suggested experimentation makes biological sense and perks our interest. Thus, the study of the effects of the LSH10/OTLD1 repressor on the patterns of *WUS* expression, although obviously beyond the scope and focus of the present paper, will be incorporated into our immediate research plans. Thanks. As for the observed phenotypes, the revised paper clearly indicates that both LSH10 loss-of-function mutant alleles did not produce overall morphological or developmental phenotypes that we could detect. Potentially, the increased expression of *WUS*, even if 30-fold higher than the wild-type basal level, still is insufficient to cause detectable phenotypes.

*Figure 4C: the complementation line pLSH10::LSH10-6xHis shows higher expression levels that of the endogenous LSH10 transcripts in wild-type background. Authors then correlate this increase of expression with a reduction in the expression of the target genes *OSR2*, *WUS*, *ABI5*. This analysis is not sufficient to state such direct correlation. Authors should present more transgenic lines, ideally a minimum of three independent transformants with different degrees of LSH10 expression. It is also known that an increase in mRNA levels can not correlate with an higher abundance of the protein. To make their conclusion solid, authors should couple expression level of LSH10 with analysis of protein level by Western Blot.*

We are a bit confused. We never intended to correlate quantitatively the "increase in expression" of LSH10 with "a reduction in the expression of the target genes". Fig. 4C simply shows the standard genetic complementation of the mutant plants, a routine approach to confirm the effect of a specific mutation. We revised the relevant Results section to clarify this point and remove any unintended quantitative

comparisons between the levels of the LSH10 expression in the mutant and complemented lines and the levels of the expression of the target genes.

Please note that it is the common standard to use genetic complementation of mutants, and we simply followed this standard. Conceptually, we see no advantage in using additional complemented transgenic lines.

Figure 6A Authors check the direct binding of LSH10 to target genes promoter. As positive binding is observed for all the gene tested, it would be nice to add also a negative control where no binding is detected.

All binding was positive because we presented the qChIP data only of those target chromatin areas that contained the DNA sequences (i.e., EMSA probes) to which LSH0 could bind in our in vitro DNA binding assays. For negative ChIP controls, we used the *ABI5* gene, but with qChIP primers away from the EMSA probes, expecting reduced or no LSH10 association with these chromatin regions. Indeed, only background-level qChIP signal was observed with these primers. These data were added to Fig. 6A of the revised paper.

I am not familiar with the use of "A.U. - Arbitrary Unit " to present ChIP enrichments, and because of this, I find the chart incomplete and hard to understand. I apologize for this lack of expertise from my side, and I thus ask if it is possible to present the ChIP data in Figure 6A as relative fold enrichment as done for Figure 6B.

The qChIP data were, in fact, calculated as "fold enrichment" as described in the original Methods ["Input Ct values were adjusted for the dilution factor and ΔCt was calculated by normalizing Ct values to the adjusted input based on the equation $\Delta Ct = Ct(\text{input}) - Ct(\text{IP})$. For anti-His6 qChIP, the % input was calculated as $\% \text{ input} = 100 \times 2^{-\Delta Ct}$; fold enrichment was calculated using % input of IP/IgG as $[\Delta Ct(\text{IP})/\Delta Ct(\text{IgG})]$ (64), and non-specific background immunosignal control, obtained with the wild-type Arabidopsis plants that do not express the His6 epitope, was subtracted from the qPCR data (28)"]. It is the subtraction of the non-specific background immunosignal that necessitated the use of "arbitrary units". However, based on the reviewer's suggestion, we did not do the subtraction and revised Fig. 6A to present the raw "fold enrichment" data as well as the non-specific signal. Thanks.

The model that the author proposes in that LSH10 is recruiter of OTLD1. However two fundamental experiments for supporting this hypothesis are missing from the current format of the work:

1) co-localization of both proteins in the same cell. The work presents evidence for direct LSH10-OTDL1 protein interaction, but for this to have a biological meaning, both proteins need to be present in the same cell (nucleus) and at the same time. Therefore expression analysis in support of this is essential (either by live imaging of - GFP tagged line for instance, or by In situ hybridization).

The reviewer is correct. However, at this time, the field does not offer the necessary molecular tools, i.e., anti-OTLD1 and anti-LSH10 antibodies suitable for in

situ immunofluorescence or transgenic plant lines with *OTLD1* and *LSH10* expressed from their native promoters and tagged with different fluorochromes. Development and validation of such tools will take at least a year, delaying publication in a very substantial and prohibitive way. Instead, we performed a detailed RT-qPCR-based analysis of the expression of the endogenous *OTLD1* and *LSH10* genes in different tissues of wild-type plants, an approach previously used to characterize the expression patterns of the *OTLD1* and *OTU1* deubiquitinase genes [Keren & Citovsky (2016) *Sci. Signal.* 9, ra125; Keren, Lacroix, Kohrman & Citovsky (2020) *iScience* 23, 100948]. The results of these experiments were presented in Fig. 3 and Table S3 of the revised paper, and they demonstrated that both genes were expressed in the same plant tissues at the same time (obviously at different levels), suggesting the availability of their protein products for functional interaction during transcriptional regulation of their target genes. Incidentally, nuclear localization of both proteins is not a prerequisite for their interaction and subsequent function; proteins often interact in the cell cytoplasm (where they, of course, are synthesized) and then are imported into the nucleus for function.

2- For LSH10 to be stated as true recruiter of OTLD1, ChIP on H2B-Ub is not sufficient. Authors should assess OTLD1 binding to the target genes in Lsh10 background. If LSH10 is a true OTLD1 recruiter, then the OTLD1 binding to the target should be abolished.

Our data demonstrate (as this reviewer kindly noted, in a "work ... rigorously conducted") that LSH10 interacts with OTLD1 in living cells, LSH10 associates with the chromatin of the OTLD1 target genes, represses their transcription and is required for deubiquitylation of H2B in the chromatin of these genes, and (as demonstrated previously) that OTLD1 is responsible for the H2B deubiquitylation of its target chromatin. Based on the Occam razor principle, the most likely reason for the absence of H2B deubiquitylation in the target chromatin in the absence of LSH10 (but in the presence of the native OTLD1) is the inability of OTLD1 to effect this deubiquitylation. The suggested experiment may be useful to confirm this idea further, but it would require inordinate amounts of time to produce transgenic *Lsh10-1/OTLD1-His6* plants and characterize/validate them before they become suitable for the ChIP experiments. This assessment is based on our direct experience with the production of double transgenic *Lsh10-1/LSH10-His6* plants, which represents close to a year of work. To avoid substantial delay in the publication of our data considered important by all three reviewers, we revised the paper (including its Title) to clarify that the recruitment of OTLD1 by LSH10 is functional recruitment and nowhere in the revised manuscript it is stated that LSH10 is a "true recruiter" in terms of strictly physical recruitment.

In this regard, we do not rule out that, even in the absence of LSH10, OTLD1 may display some (albeit non-functional) chromatin association due to other members of the LSH10/OTLD1 repressor complex, such as the SUVR5 histone methyltransferase [Krichevsky, Gutgarts, Kozlovsky, Tzfira, Sutton, Sternglanz, Mandel & Citovsky (2007) *Dev. Biol.* 303, 259-269], the KDM1C histone lysine demethylase [Krichevsky, Zaltsman, Lacroix & Citovsky (2011) *Proc. Natl. Acad. Sci. USA* 108, 11157-11162; Keren, Lapidot & Citovsky (2019) *Epigenetics* 14, 602-610],

histone acyltransferases and others. Thus, the proposed experiment may not provide the ultimate proof for recruitment.

Minor comments:

The introduction would benefit with the addition of some background information on what histone ubiquitination/deubiquitination is and the roles of this modification in transcription.

The original Introduction already explained and referenced that "OTLD1 mainly functions as a transcriptional co-repressor ... by associating with the target chromatin and deubiquitylating histone 2B (H2B) at the occupied regions, thereby promoting the erasing or writing of euchromatic histone acetylation and methylation marks". However, as this reviewer requested, we expanded on this explanation in the revised paper.

When analyzing the LSH10 protein structure, authors could add a 3D model, as the one reported here <https://www.uniprot.org/uniprotkb/Q9S7R3/entry>

We already demonstrated the 3D model of LSH10, as predicted by AlphaFold (AF-Q9S7R3), and compared it to the DNA binding domain of the Cre recombinase. These data are presented in Fig. S4 and discussed in the paper. Note that the 3D model predicted by UniProt as suggested by the reviewer is very similar to the model predicted by AlphaFold as shown in Fig. S4. Thanks.

*Figure 4B: expression analysis of OTLD1 targets expression in *Ish10* mutant backgrounds. It would be useful to have also the expression level of such genes in the *otld1* mutant, to directly compare the expression levels in the two genetic backgrounds.*

*Figure 6B: ChIP of H2B-Ub in *Ish10* background. It would be useful to add also here the degree of H2B-Ub enrichment in *otld1* mutant background.*

The effect of *OTLD1* gain- and loss-of-function mutants on the target gene expression as well as H2B deubiquitylation by *OTLD1* was already described in detail by Keren & Citovsky (2016) *Sci. Signal.* 9, ra125; repeating these experiments would unnecessarily burden the paper with previously published data. Furthermore, because the lack of the *OTLD1* activity most likely is compensated by one or more of its 12 OTU protein family homologs encoded by the Arabidopsis genome (Keren & Citovsky, 2016), the proposed experiments are not feasible technically and conceptually. This point was clarified in the revised paper.

Line 52: "To achieve this regulation, histone-modifying enzymes are thought to function in complexes with transcription factors that contain DNA-binding domains and, thus, can provide the DNA binding capacity to the histone modifier-transcription factor complex and thereby recruit histone-modifying enzymes to the target promoters.

I find this sentence very wordy and I had to read it multiple times to understand it properly. Authors could break this into two sentences to make it clearer.

This long and complex sentence was reworded to break it into two shorter and clearer sentences as suggested. Thanks.

Line 55: "Indeed, in plants, different transcription factors have been shown to recruit such diverse histone modifiers as histone methyltransferases, histone acetyltransferases, histone demethylases, and Polycomb repressive complexes that promote histone trimethylation and monoubiquitylation". Please provide some references in support of this sentence.

We felt that this short sentence in the Introduction that simply lists the histone modifiers that interact with transcription factors did not require references because the same interactions were discussed in detail and fully referenced in the Discussion. However, as this reviewer requested, references were added also to this statement in the Introduction.

Reviewers' comments:

Reviewer #1 (Remarks to the Author):

all my criticisms and suggestions have been addressed

Reviewer #3 (Remarks to the Author):

I thank all the authors for their thorough response to my comments, and for having taken on board some of my suggestions to improve their work.

I do, however, still have some reservations for a couple of points:

1. Figure 4D (ex 4C). I apologize for having misunderstood the message that the authors wanted to give with this analysis. I am perfectly aware of the common standard rules to assess functionality of tagged proteins. I was confused by the fact that authors perform statistical analysis as to compare WT to their transgenics and they comment as such in their manuscript "Line 214 -Collectively, these observations indicate that LSH10 acts as a transcriptional repressor of most of the known OTLD1 target genes.". It could be clearer if the authors could add a sentence such "this analysis confirms that our tagged version is able to complement the mutant".

2. Co-localization of LSH10 and OTLD1. I understand that creating new marker lines is a task that requires months of work and waiting time, which is not compatible with the time allocated for a manuscript revision. Instead, the authors performed analysis of transcripts levels in different tissues. Although I understand the point of view of the authors and their good intentions, I do not agree that this new analysis resolves the issue about the two proteins co-localizing in the same cell. All the analysis brought to show OTLD1-LSH10 functional interaction indirectly supports that they work together in the nucleus. Direct proof still lacks from the revised version. As referring to their comment "Incidentally, nuclear localization of both proteins is not a prerequisite for their interaction and subsequent function; proteins often interact in the cell cytoplasm (where they, of course, are synthesized) and then are imported into the nucleus for function". This statement is correct, and indeed co-localization in the nucleus does not mean direct interaction. An in-situ FRET-FLIM or Co-IPs, expressing tagged proteins under their native promoters, would resolve this issue. I simply did not ask for such experiments (that need a lot of time for the lines to be generated) because the authors bring enough solid data to support a direct interaction at the protein level.

3. ChIP of OTLD1 in Lsh10 mutant background to confirm LSH10 as direct recruiter. As for point 2 above, I do understand the time and effort required to obtain marker lines and introgress them in different backgrounds. However, the lack of time should not be used as justification to compromise on quality. I thus stay on my opinion: to confirm that LSH10 is a recruiter of OTLD1, ChIP on OTLD1 in Lsh10 mutant background are needed. Without this analysis, the title and the manuscript, although modified to "functional recruitment" are still misleading and convey a message that it is not supported by experiments, therefore inflating the findings. The issue here is not "functional", but is "recruitment", which implies a mechanism of indispensability of one protein (LSH10) to another (OTLD1), for having the latter to function correctly. I find the use of the Occam's razor principle to support their statements rather unprofessional. Occam's razor is a sort of tool that might be useful when developing new hypothesis. However, it is based on the assumption that the simplest explanation is the correct one, which can be also wrong (and proved wrong many times in several past examples). Authors can use the Occam's razor principle to speculate on their findings in the Discussion section, and leave the readers to agree/disagree with it based on the results they bring.

Reviewer 3

We thank the reviewer for the additional suggestions, all of which have been addressed in the revised paper as described below.

1. Figure 4D (ex 4C). I apologize for having misunderstood the message that the authors wanted to give with this analysis. I am perfectly aware of the common standard rules to assess functionality of tagged proteins. I was confused by the fact that authors perform statistical analysis as to compare WT to their transgenics and they comment as such in their manuscript "Line 214 -Collectively, these observations indicate that LSH10 acts as a transcriptional repressor of most of the known OTLD1 target genes.". It could be clearer if the authors could add a sentence such "this analysis confirms that our tagged version is able to complement the mutant".

Excellent suggestion. We rewrote the corresponding paragraph to include the suggested clarification. Thanks.

2. Co-localization of LSH10 and OTLD1. I understand that creating new marker lines is a task that requires months of work and waiting time, which is not compatible with the time allocated for a manuscript revision. Instead, the authors performed analysis of transcripts levels in different tissues. Although I understand the point of view of the authors and their good intentions, I do not agree that this new analysis resolve the issue about the two proteins co-localizing in the same cell. All the analysis brought to show OTLD1-LSH10 functional interaction indirectly support that they work together in the nucleus. Direct proof still lacks from the revised version. As referring to their comment " Incidentally, nuclear localization of both proteins is not a prerequisite for their interaction and subsequent function; proteins often interact in the cell cytoplasm (where they, of course, are synthesized) and then are imported into the nucleus for function". This statement is correct, and indeed co-localization in the nucleus does not mean direct interaction. An in-situ FRET-FLIM or Co-IPs, expressing tagged proteins under their native promoters, would resolve this issue. I simply did not ask for such experiments (that need a lot of time for the lines to be generated) because the authors bring enough solid data to support a direct interaction at the protein level.

We thank this reviewer for the realistic assessment of the time and effort needed for further in situ studies using FRET-FILM and ChIP, and for the appreciation of the interaction data. Also, we are grateful for understanding that the colocalization "requires months of work and waiting time, which is not compatible with the time allocated for a manuscript revision". Obviously, we also agree that "colocalization in the nucleus does not mean direct interaction"; this is why we used BiFC (direct interaction) and FRET (exceedingly close colocalization). As for colocalization in the same cell, we believe that our biological data linking LSH10 and OTLD1 as well as their overlapping patterns of expression in different tissues/organs provide a sufficient basis for our model of the coordinate action of these proteins on their target genes. Obviously direct colocalization of LSH10 and OTLD1 in the same cell using

fluorescently tagged plant lines would further support this model. This point was clarified in the revised paper.

3. ChIP of OTLD1 in Lsh10 mutant background to confirm LSH10 as direct recruiter. As for point 2 above, I do understand the time and effort required to obtain marker lines and introgress them in different backgrounds. However, the lack of time should be not use as justification to compromise on quality. I thus stay on my opinion: to confirm that LSH10 is a recruiter of OTLD1, ChIP on OTLD1 in Lsh10 mutant background are needed. Without this analysis, the title and the manuscript, although modified to "functional recruitment" are still misleading and convey a message that it is not supported by experiments, therefore inflating the findings. The issue here is not "functional", but is "recruitment", which implies a mechanism of indispensability of one protein (LSH10) to another (OTLD1), for having the latter to function correctly. I find the use of the Occam's razor principle to support their statements rather unprofessional. Occam's razor is a sort of tool that might be useful when developing new hypothesis. However, it is based on the assumption that the simplest explanation is the correct one, which can be also wrong (and proved wrong many times in several past examples). Authors can use the Occam's razor principle to speculate on their findings in the Discussion section, and leave the readers to agree/disagree with it based on the results they bring.

Overall, we agree with the reviewer that the suggested experiments could provide more definitive proof of recruitment (the key word here is "could" as our work focuses on functional recruitment and not necessarily on the physical presence of OTLD1 at the target chromatin which, in the absence of LSH10, could be brought there by other transcription factors in a non-functional manner). Also, we would like to point out that no manuscript can present a totally complete story, and one of the main goals of a good research paper is not only to report data but also to provide a foundation for future studies.

Yet, we completely disagree with the reviewer's statement that "without this analysis ... the manuscript ... [is] misleading and convey[s] a message that it is not supported by experiments". In fact, we (and apparently two other reviewers) think that all data in this paper support the recruitment model.

To solve our differences in opinion with this reviewer, we propose the following three changes to the manuscript that directly address the reviewer's criticism of "inflating the finding":

- (1) We revised the paper's title to avoid the term "recruitment" altogether;
- (2) as the reviewer suggested, we revised the entire manuscript "to speculate on ... [the notion of recruitment mostly] in the Discussion section, and leave the readers to agree/disagree with it based on the results";
- (3) we revised Discussion to include a "limitation of the study"-type statement that "the future examination of this notion of LSH10-mediated recruitment of OTLD1 to the target chromatin will benefit from a demonstration of the physical absence of OTLD1 from the target chromatin in the Lsh10 mutant plants".